# SynCL: A Synergistic Training Strategy with Instance-Aware Contrastive Learning for End-to-End Multi-Camera 3D Tracking

**Shubo Lin**[1,2,3], **Yutong Kou**[1,2,3], **Zirui Wu**[6], **Shaoru Wang**[1]
**Bing Li**[1,3,5], **Weiming Hu**[1,2,3,4], **Jin Gao**[1,2,3*]
[1]State Key Laboratory of Multimodal Artificial Intelligence Systems (MAIS), CASIA
[2]School of Artificial Intelligence, University of Chinese Academy of Sciences
[3]Beijing Key Laboratory of Super Intelligent Security of Multi-Modal Information
[4]School of Information Science and Technology, Shanghai Tech University
[5]People AI, Inc, [6]Harbin Institute of Technology
{linshubo2023}@ia.ac.cn, {jin.gao}@nlpr.ia.ac.cn

## Abstract

While existing query-based 3D end-to-end visual trackers integrate detection and tracking via the *tracking-by-attention* paradigm, these two chicken-and-egg tasks encounter optimization difficulties when sharing the same parameters. Our findings reveal that these difficulties arise due to two inherent constraints on the self-attention mechanism, i.e., over-deduplication for object queries and self-centric attention for track queries. In contrast, removing the self-attention mechanism not only minimally impacts regression predictions of the tracker, but also tends to generate more latent candidate boxes. Based on these analyses, we present SynCL, a novel plug-and-play synergistic training strategy designed to co-facilitate multi-task learning for detection and tracking. Specifically, we propose a Task-specific Hybrid Matching module for a weight-shared cross-attention-based decoder that matches the targets of track queries with multiple object queries to exploit promising candidates overlooked by the self-attention mechanism and the bipartite matching. To flexibly select optimal candidates for the one-to-many matching, we also design a Dynamic Query Filtering module controlled by model training status. Moreover, we introduce Instance-aware Contrastive Learning to break through the barrier of self-centric attention for track queries, effectively bridging the gap between detection and tracking. Without additional inference costs, SynCL consistently delivers improvements in various benchmarks and achieves state-of-the-art performance with 58.9% AMOTA on the nuScenes dataset. Code and raw results are available at `https://github.com/shubolin028/SynCL`.

## 1 Introduction

The perception system is an indispensable component in autonomous driving. Within the system, accurate 3D multi-object tracking (MOT) provides reliable observations for planning. Camera-based tracking algorithms have raised significant attention due to their cost-effectiveness [1]. Traditional trackers following the *tracking-by-detection* paradigm often suffer from tedious and unstable post-processing in new scenarios [2, 3]. To achieve cross-domain adaptability, the prevailing methods [4–8] adopt the *tracking-by-attention* paradigm, which integrates single-frame detection and inter-temporal tracking by encoding new-born and persistent targets as object queries and track queries respectively

---

[*]Corresponding author.

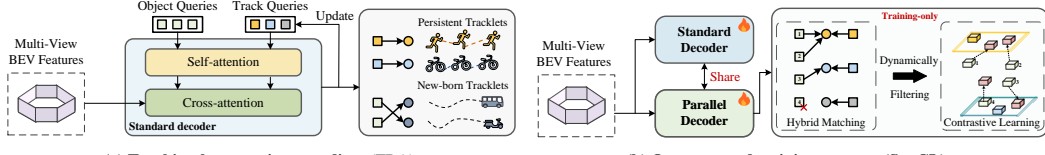

**(a) Tracking-by-attention paradigm** (TBA)     **(b) Our proposed training strategy** (SynCL)

Figure 1: Comparisons between the *tracking-by-attention* paradigm and our proposed plug-and-play training strategy, SynCL. SynCL consists of Task-specific Hybrid Matching, Instance-aware Contrastive Learning powered by a Dynamic Query Filtering module.

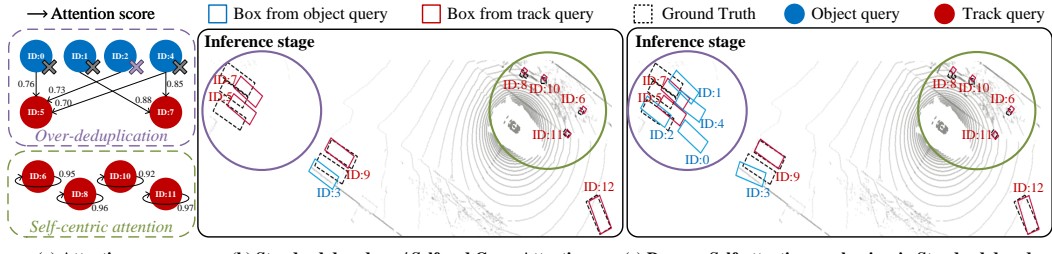

**(a) Attention map**     **(b) Standard decoder w/ Self and Cross Attention**     **(c) Remove Self-attention mechanism in Standard decoder**

Figure 2: We compare results from the standard decoder in Fig. 2b with those from the decoder without self-attention in Fig. 2c in inference stage. The self-attention mechanism exhibits over-deduplication for object queries and self-centric attention for track queries. Results are from the model trained without utilizing SynCL.

(Fig. 1a). However, the optimization of detection and tracking, two interdependent yet distinctly characterized tasks, remains an open question due to the sharing of the same model parameters within the end-to-end training pipeline [9, 10]. Previous works attribute the optimization difficulties to conflicts in representation [6] and gradient flows across distinct queries [8]. DQTrack [11] attempts to train detection and learning-based association separately, but this two-stage training pipeline fails to enable synergistic representation learning. ADATrack [6] introduces an additional attention-based association module to strengthen inter-query connections, yet consequently increases model complexity. OneTrack [8] groups object queries and blocks conflicting gradient flows, but its design is confined to a specific detector, thus lacking generalizability. A question is: *Is there a universal training strategy that can effectively address the optimization difficulties of detection and tracking under shared model parameters through a joint training paradigm without affecting inference speed?*

To seek the answer, we first conduct a comprehensive analysis of attention mechanisms in the *tracking-by-attention* paradigm. We find that the self-attention mechanism exhibits two inconspicuous characteristics for object queries and track queries, respectively:

- Over-deduplication for object queries: Referring to the purple dashed box in Fig. 2a, the self-attention mechanism in the decoder assigns high attention scores from object queries to similar track queries. Compared to the prediction results before and after removing the self-attention mechanism (see purple circles in Fig. 2b and Fig. 2c), we observe that these high attention scores effectively suppress duplicate predictions (e.g., ID 0, ID 1 and ID 4). Although this de-duplication plays a similar role of Non-Maximum Suppression (NMS) to some extent, it inadvertently results in information loss, potentially eliminating some high-quality candidates (e.g., ID 2).

- Self-centric attention for track queries: Referring to the green dashed box in Fig. 2a, track queries exhibit high attention scores towards themselves, lacking interaction with object queries. As illustrated in the green circles of Fig. 2b and Fig. 2c, the imprecise predictions of track queries (e.g., ID 6, ID 8, etc.) change little before and after the removal of the self-attention mechanism.

Despite the above issues, directly replacing the self-attention mechanism with NMS will lead to the collapse of joint detection and tracking training. In another way, optimization difficulties may be mitigated by a weight-shared cross-attention-based parallel decoder, following [12]. However, simultaneously addressing the over-deduplication of object queries and the self-centric issues of track queries within a parallel decoder remains non-trivial (see Appendix B.3 for more details).

To thoroughly tackle these challenges, we propose a novel synergistic training strategy for the two tasks based on the *tracking-by-attention* paradigm, which we term SynCL. As presented in Fig. 1b, SynCL introduces a weight-shared parallel decoder during training, which removes the

self-attention mechanism. Concretely, we propose a **Task-specific Hybrid Matching** module that additionally employs one-to-many assignment for object queries to uncover and preserve promising candidates overlooked by the self-attention mechanism in the parallel decoder. With more diverse and refined candidate representations from one-to-many label supervision, predictions from object queries are robust to over-deduplication. Additionally, we design a **Dynamic Query Filtering** module based on the Gaussian Mixture Model (GMM) that flexibly selects reliable object queries for the one-to-many assignment according to the prediction quality and optimization state of the model. Moreover, to get rid of constraints from self-centric attention on track queries and establish cross-task connections, we introduce **Instance-aware Contrastive Learning** that aligns joint object and track queries corresponding to identical ground truth targets while separating irrelevant queries in the latent space and thus provides synergistic learning with richer and quality features.

In summary, our main contributions are as follows: **(1)** We reveal that the optimization difficulties of joint detection and tracking training lie in the imperceptible effect of the self-attention mechanism across different queries. **(2)** We propose SynCL, a synergistic training strategy compatible with any *tracking-by-attention* paradigm tracker to address multi-task learning challenges. SynCL implements dynamic filtering-based hybrid matching and instance-aware contrastive learning to improve the performance of both detection and tracking. **(3)** We empirically show that SynCL brings remarkable improvements over various *tracking-by-attention* baselines with acceptable extra training cost and establishes new state-of-the-art performance on the camera-based nuScenes MOT benchmark.

## 2 Related Work

**Tracking-by-detection.** Traditional tracking methods in 2D or 3D [2, 3, 13, 14] follow the *Tracking-by-detection* paradigm, which decouples detection and tracking, regarding tracking as a post-processing step. Early works associate targets with existing trajectories by handcrafted features such as appearance similarity [1], geometric distance [15–18]. *Tracking-by-detection* paradigm achieves further advancements when combined with learning-based association methods. Specifically, learning modules such as GNN [19, 20] and Transformers [21, 11] are employed to merge multifaceted information. However, detection and tracking are complementary yet non-independent tasks. This partitioned paradigm design does not effectively optimize both tasks simultaneously.

**Multi-task learning in MOT.** Building upon DETR [22] of encoding detection targets as object queries, MOTR [23] utilizes the track queries, enriched with prior features of object queries, to propagate across frames, enabling an end-to-end tracking framework. MUTR3D [4] and PF-Track [5] extend the end-to-end framework to multi-camera 3D MOT. However, sharing the same model parameters for detection and tracking within an end-to-end framework introduces optimization difficulties in multi-task learning. Subsequent works [9–11] attempt to address these difficulties. STAR-Track [24] introduces a Latent Motion Model (LMM) to account for changes in query appearance features across frames and ADA-Track [6] integrates the appearance and geometric clues of the track queries to refine object queries. Both of them increase inference time. TQDTrack [25] introduces temporal query denoising to enhance the modeling of track queries. OneTrack [8] utilizes attention mask to block conflicting gradient flows, but its design is confined to [26]. Despite these efforts, a general strategy for optimization difficulties in joint detection and tracking training remains unsolved.

## 3 Methodology

This section elaborates on our proposed method, SynCL, for multi-view 3D object tracking. We start with an overview of *tracking-by-attention* paradigm trackers in Sec. 3.1 and an analysis to validate the constraints on the self-attention mechanism in Sec. 3.2. Then, we provide a detailed construction of hybrid matching for the parallel decoder in Sec. 3.3 and introduce the filtering module design for one-to-many assignment in Sec. 3.4. To achieve synergistic end-to-end learning for detection and tracking, we further propose instance-aware learning in Sec. 3.5.

### 3.1 Preliminaries

SynCL is designed to be integrated into any *tracking-by-attention* paradigm multi-view 3D tracker. These trackers typically consist of a convolutional network-based feature encoder (*e.g.*, ResNet [27], VoVNet [28]), a transformer-based decoder, and a predictor head for object classes and boxes. The

goal of 3D MOT is to generate detection boxes of the instances in the perceptual scene with consistent IDs. Given $N$ images $\mathbf{I}_t = \{\mathbf{I}_t^i, i = 1, 2, \ldots, N\}$ captured by surrounding cameras at timestamp $t$, the encoder first extracts image features $\mathbf{F}_t = \{\mathbf{F}_t^i, i = 1, 2, \ldots, N\}$ with 3D point position embeddings [29, 30]. A crucial aspect of realizing end-to-end tracking is to allow the prior latent encoding from the previous frame, track queries $\mathbf{Q}_{t-1}^{trk}$, to be updated iteratively by the decoder interacting with the current frame image features. To detect new-born targets, object queries $\mathbf{Q}^{obj}$ with a fixed number of $\mathbf{N}_{obj}$ are merged together and fed into the decoder:

$$\mathbf{Q}_t^{trk} = \text{Decoder}(\mathbf{F}_t, \mathbf{Q}_{t-1}^{trk} \cup \mathbf{Q}^{obj}) \tag{1}$$

Each query $q \in \{\mathbf{Q}^{trk}, \mathbf{Q}^{obj}\}$ includes an embedding vector $e$, assigned with a unique 3D reference $c$, i.e., $q = \{e, c\}$. In the center-point anchor-free predictor, the reference points are actively involved in the parsing of the bounding box:

$$\mathbf{B}_t = \text{Box}(e_t, c_t), \quad \mathbf{P}_t = \text{Classification}(e_t) \tag{2}$$

where $\mathbf{B}_t = \{b^1, b^2, ..., b^n\}$ is the set of prediction 3D boxes and $\mathbf{P}_t = \{p^1, p^2, ..., p^n\}$ is the set of classification scores for all categories ($\mathbf{N}_c$). Each box ($b$) consists of 3D center point ($\widetilde{c} \in \mathbb{R}^3$), 3D box size ($s \in \mathbb{R}^3$), yaw angle ($\theta \in \mathbb{R}^1$), and BEV velocity ($v \in \mathbb{R}^2$).

Considering the heterogeneity of the two tasks, object queries and track queries are supervised with different label assignment strategies. Object queries are precisely matched with new-born targets via the Hungarian algorithm and assigned unique IDs, while track queries are directly mapped to persistent targets with consistent IDs.

However, matching object queries to new-born objects contradicts the detection pre-training, substantially weakening the supervision signals for object queries. Consequently, when track queries are initialized with object queries from the previous frame, these low-quality priors inevitably exacerbate the optimization difficulties in inter-temporal tracking. We find that this suboptimal assignment stems from the inconspicuous nature of the self-attention mechanism and the bipartite matching, preventing synergistic training for both identity-agnostic detection and identity-aware tracking.

## 3.2 Analysis

To further validate our findings in Fig. 2, we visualize the self-attention distributions from shallow to deep decoder layers. As shown in Fig. 3, over-deduplication stems from the attention of object queries to track queries, which evolves from global non-differentiated responses to local similarity responses. For track queries, self-centric attention prevents them from collaborative learning with object queries. To this end, we separate the self-attention mechanism and resolve these issues through a divide-and-conquer method in the weight-shared parallel decoder.

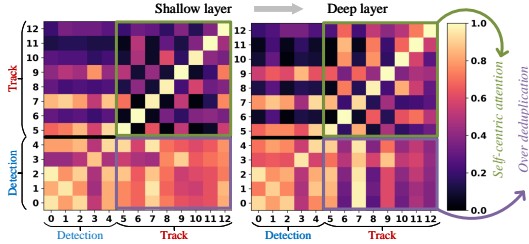

Figure 3: Analysis of self-attention heatmap in the standard decoder. The annotation numbers of the heatmap are aligned with the ID numbers in Fig. 2.

## 3.3 Task-specific Hybrid Matching

Based on the regression characteristics of cross-attention, we first construct a weight-shared parallel decoder with the removal of the self-attention mechanism. To further enhance the inter-temporal modeling capability and exploit promising candidates, we implement one-to-one and one-to-many label assignment for track queries and object queries, respectively.

**Parallel Decoders.** We design two parallel decoders: a standard decoder (S-decoder) and a cross-attention-based decoder (C-decoder). A block in the standard decoder consists of a self-attention layer, a cross-attention layer and a feed-forward network (FFN). The embeddings of both object and track queries are concatenated and fed into the two decoders, denoted as $e_s^l$ and $e_c^l$, where $e^l = e_{obj}^l \cup e_{trk}^l$. The superscript $l = \{0, 1, ..., 5\}$ indicates the index of the decoder block and we omit the notation of the frame index $t$ for simplicity. The initial inputs to both decoders are identical, i.e., $e_s^0 = e_c^0$. In the $l_{th}$ block of the S-decoder, the state of both queries is first updated through the self-attention layer

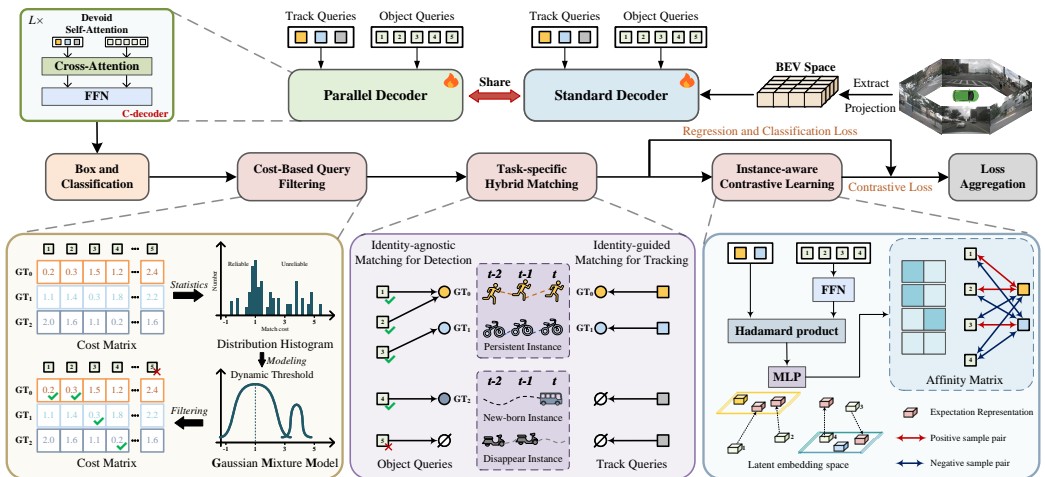

Figure 4: **Overview of SynCL.** SynCL is based on *tracking-by-attention* paradigm trackers, with two weight-shared parallel decoders: a S-decoder (standard decoder) and a C-decoder (devoid of self-attention layers). In C-decoder, hybrid matching with one-to-many and one-to-one assignment is applied for object queries and track queries, respectively. Besides, a dynamic filtering module is designed to flexibly select reliable object queries for the one-to-many assignment. With identical ground-truth matching, contrastive learning unifies the representations between object and track queries, co-facilitating multi-task learning for detection and tracking.

bidirectionally, which is formulated as:

$$\hat{e}_s^l = e_s^l + \text{Attention}_{\text{self}}(Query = Key = Value = e_s^l) \tag{3}$$

where $\text{Attention}_{\text{self}}(\cdot)$ indicates the self-attention layer. Next, the two types of queries are interacted with multi-view image features, where object queries generate potential candidates through the positional priors, and track queries locate persistent targets utilizing representation prior:

$$e_s^{l+1} = \text{FFN}(\hat{e}_s^l + \text{Attention}_{\text{cross}}(Query = \hat{e}_s^l, \ Key = Value = \mathbf{F})) \tag{4}$$

where $\text{Attention}_{\text{cross}}(\cdot)$ indicates the cross-attention layer. The C-decoder retains the cross-attention layers and FFNs, sharing parameters with the S-decoder. In this setup, the cross-attention layers take image features as the sole source set (key and value), dynamically aggregating and updating both queries in the stacked decoder blocks:

$$e_c^{l+1} = \text{FFN}(e_c^l + \text{Attention}_{\text{cross}}(Query = e_c^l, \ Key = Value = \mathbf{F})) \tag{5}$$

Due to the absence of mutual interaction between object and track queries, it is challenging for the C-decoder to remove duplicate candidates with proximal or occluded targets. Thus, we design distinct label assignment tailored to the characteristics of the two parallel decoders and the specific requirements of their respective tasks.

**Hybrid Matching.** End-to-end *tracking-by-attention* paradigm trackers rely on one-to-one assignment for both object and track queries. In the S-decoder, assignment of object queries $\sigma_s^{obj}$ can be obtained by performing the Hungarian algorithm between predictions and new-born targets, while assignment of track queries $\sigma_s^{trk}$ is consistent with the assigned IDs, which is formulated as:

$$\sigma_s^{obj} = \arg\min_{\sigma} \sum_{i=1}^{M} \mathcal{C}_{match}\left(b_{\sigma(i)}^{obj}, p_{\sigma(i)}^{obj}, g_{new}^i\right) \tag{6}$$

$$\sigma_s^{trk} = \left\{\sigma \mid \text{ID}_{\sigma(i)}^{trk} = \hat{\text{ID}}_i, \ \hat{\text{ID}} \in g_{pst}, i = 1, 2, \ldots, N\right\} \tag{7}$$

where the $g_{new}$ and $g_{pst}$ are $M$ new-born and $N$ persistent ground truths (GTs). $\mathcal{C}_{match}$ is the matching cost between predictions and GTs. $\sigma(\cdot)$ is the optimal permutation of $M$ or $N$ indices, where $k_{th}$ ground truth target is assigned to $\sigma(k)_{th}$ prediction. Each query corresponds to a ground-truth target and vice versa. The track queries corresponding to disappearing targets are not assigned labels. Following [30, 31], we utilize a combination of the classification score and the 3D box as the matching cost metric, which is formulated as:

$$C_{match}(b, p, g) = \text{FocalLoss}(p, \hat{p}) + \text{L1}(b, \hat{b}) \tag{8}$$

In the C-decoder, we maintain the identity-guided principle for track queries to avoid ambiguity. However, lacking the NMS effect of self-attention, object queries in C-decoder tend to output multiple similarity predictions. To improve the quality of candidate boxes and boost the detection of lost targets, we perform the one-to-many assignment for object queries in C-decoder with persistent GTs, i.e., one ground-truth target corresponds to multiple object queries:

$$
\sigma_c^{obj} = \left\{ \arg\min_\sigma \sum_{j=1}^{K} \mathcal{C}_{match} \left( b_{\sigma_i(j)}^{obj}, p_{\sigma_i(j)}^{obj}, g_{pst}^i \right) \right\}_{i=1}^{N} \tag{9}
$$

where the $K$ is the maximum assignment constraint per GT for object queries. In order to obtain more positive sample pairs for contrastive learning, we do not match object queries with all ground-truth targets. In Sec. 4.3, we explore the impact of different matching variants in detail.

## 3.4 Dynamic Query Filtering

To effectively explore more high-quality candidate samples for the identity-agnostic matching in Hybrid Matching, we propose a cost-based filtering module with a dynamic threshold $\delta$, which filters out unreliable matching pairs whose cost exceeds $\delta$. Specifically, we use Gaussian Mixture Model (GMM) to unsupervisedly cluster matching sample pairs into reliable/unreliable clusters based on their matching cost $z$, which is formulated as:

$$
\boldsymbol{\pi}, \boldsymbol{\mu}, \boldsymbol{\Sigma} = \arg\min \sum_z - \log \sum_{i=1}^{N_c} \pi_i \mathcal{N}(z|\mu_i, \Sigma_i) \tag{10}
$$

where $\mathcal{N}(z|\mu_i, \Sigma_i)$ is the Gaussian probability density function with mean $\mu_i$ and covariance $\Sigma_i$, and $\pi_i$ is the weight for the $i$-th component, satisfying $\sum_{i=1}^{2} \pi_i = 1$. The cluster with lower $\mu_i$ is considered as the reliable cluster and we use its mean $\mu_{reliable} = \min_i \mu_i$ as the dynamic threshold $\delta$. Notably, this dynamic design intelligently constructs a progressive learning schedule ranging from simple to complex depending on the optimization status of the model.

## 3.5 Instance-aware Contrastive Learning

To further transfer object queries' knowledge of persistent targets learned by one-to-many assignment in Hybrid Matching to track queries, we employ contrastive learning for object queries and their corresponding track queries with identical ground-truth matching. As shown in Fig. 4, contrastive learning pulls representations of object queries closer to those of track queries assigned to the same instance, while pushing them further apart from representations of distinct instances in the latent space. Unlike self-supervised learning methods [32, 33], which involve a large number of sample pairs (e.g., 4096), there are typically fewer than 100 track queries propagated from the previous frame to serve as samples pairs. To alleviate this issue of insufficient sample pairs, we draw on [34] and utilize kernel-based contrastive learning, which is formulated as:

$$
\mathcal{L}_{CL}(\mathcal{K}; \tau) = -\log \frac{\exp(\mathcal{K}[e_i^{obj}, e_j^{trk}]/\tau)}{\sum_{k=1, k \neq j}^{N} \exp(\mathcal{K}[e_i^{obj}, e_k^{trk}]/\tau)} \tag{11}
$$

where $e^{obj}$ and $e^{trk}$ are the embeddings of object and track queries in the C-decoder. $\tau$ is a temperature hyper-parameter [35]. $\mathcal{K}[\cdot]$ refers to the kernel function mapping with $\phi(\cdot)$ and $g_\theta(\cdot)$:

$$
\mathcal{K}[e^{obj}, e^{trk}] = \left\langle \phi(g_{\theta_{obj}}(e^{obj})), \phi(g_{\theta_{trk}}(e^{trk})) \right\rangle \tag{12}
$$

where $\langle \cdot, \cdot \rangle$ is the inner product. When $\phi(\cdot)$ is a linear mapping function, $\mathcal{K}[\cdot]$ can be converted to:

$$
\mathcal{K}[e^{obj}, e^{trk}] = \sum \left\langle \phi(g_{\theta_{obj}}(e^{obj}), g_{\theta_{trk}}(e^{trk})) \right\rangle_{\mathcal{H}} \tag{13}
$$

where $\langle \cdot, \cdot \rangle_{\mathcal{H}}$ is the Hadamard product. In practice, we employ $\phi(\cdot)$ and $g_{\theta_{obj}}(\cdot)$ as a multi-layer perceptron (MLP) and a FFN. We do not utilize any projection for track queries for simplicity:

$$
\mathcal{K}[e^{obj}, e^{trk}] = \mathrm{MLP} \left( \left\langle (\mathrm{FFN}(e^{obj}), e^{trk}) \right\rangle_{\mathcal{H}} \right) \tag{14}
$$

We apply Instance-aware Contrastive Learning only in the training stage. During inference, the additional model parameters used for contrastive learning from Eq. 14 are discarded.

Table 1: Comparison results of adopting SynCL with *tracking-by-attention* (TBA) baselines on nuScenes *val* set. We divide the experiment into four groups based on the detector settings. * denotes results from [6]

| Method | Backbone | Detector | Resolution | Tracking | | | | | | | Detection | |
|---|---|---|---|---|---|---|---|---|---|---|---|---|
| | | | | AMOTA | AMOTP↓ | Recall | MOTA | IDS↓ | FP↓ | FN↓ | NDS | mAP |
| MUTR3D* [4] | R101 | DETR3D | $900 \times 1600$ | 32.1% | 1.448 | 45.2% | 28.3% | **474** | 15269 | 43828 | - | - |
| SynCL (ours) | R101 | DETR3D | $900 \times 1600$ | **35.8%** | **1.391** | **49.2%** | **32.9%** | 588 | **14311** | **40740** | - | - |
| PF-Track [5] | V2-99 | PETR | $320 \times 800$ | 40.8% | 1.343 | 50.7% | 37.6% | **166** | **15288** | 40398 | 47.7% | 37.8% |
| SynCL (ours) | V2-99 | PETR | $320 \times 800$ | **44.7%** | **1.262** | **56.5%** | **40.8%** | 203 | 15344 | **36801** | **49.7%** | **39.6%** |
| Baseline#1 [5] | V2-99 | PETRv2 | $320 \times 800$ | 43.2% | 1.272 | 55.0% | 40.6% | **173** | 14106 | 37065 | 50.4% | 41.0% |
| SynCL (ours) | V2-99 | PETRv2 | $320 \times 800$ | **45.7%** | **1.260** | **56.8%** | **43.0%** | 170 | **13411** | **36756** | **51.1%** | **42.0%** |
| Baseline#2 [4] | V2-99 | Stream | $320 \times 800$ | 49.6% | 1.164 | 57.3% | 42.9% | **411** | 13962 | 33526 | 57.6% | 48.5% |
| SynCL (ours) | V2-99 | Stream | $320 \times 800$ | **51.8%** | **1.149** | **58.8%** | **45.2%** | 540 | **13639** | **33368** | **58.7%** | **49.2%** |

## 4 Experiments

### 4.1 Experimental Setup

**Datasets and Evaluation Metrics.** We evaluate the proposed SynCL on nuScenes [36], a large-scale benchmark for autonomous driving, which contains 700, 150, and 150 scenes for training, validation, and testing, respectively. The dataset covers 10 types of common objects on the road. For the tracking task, nuScenes selects a subset of 7 movable categories, excluding static objects such as traffic cones. Consistent with official evaluation metrics for 3D MOT, we primarily report AMOTA (average multi-object tracking accuracy) and AMOTP (average multi-object tracking precision) [37]. We also report metrics of NDS (nuScenes detection score) and mAP (mean Average Precision) for detection to fully evaluate multi-task performance.

**Baseline and Implementation Details.** To assess the generality of our proposed synergistic training strategy, we integrate SynCL within PF-Track [5] and MUTR3D [4] frameworks using different detectors. Specifically, our experiments involve four query-based detectors: DETR3D [30], PETR [29], PETRv2 [31] and StreamPETR [26]. For DETR3D and PETR, we follow the experimental setups in ADA-Track [6] and PF-Track, respectively. Regarding PETRv2, we establish the baseline by substituting only the detector in the PF-Track framework referred to as **Baseline#1**. Unlike One-track [8], our experiments with the StreamPETR detector in MUTR3D framework are initialized from pre-trained detector weights and we then train the tracker in the sliding window mode to ensure the temporal gradient flow, denoted as **Baseline#2**. Following common practice, we set the training epoch to 24 for experiments with resolution of $900/640 \times 1600$, while we set it to 12 for experiments with resolution of $320 \times 800$. All methods are trained using the AdamW [38] optimizer with a weight decay of $1.0 \times 10^{-2}$. The learning rate is initially set to $2.0 \times 10^{-4}$ and following a cosine annealing schedule. We conduct all experiments on eight A100 GPUs with a batch size of 1 and each training batch consists of a clip of three consecutive frames from different timestamps. For the ablation studies, configurations of PF-Track with a small resolution $320 \times 800$ are employed as the default.

**Inference.** We claim that SynCL does not employ the weight-shared parallel decoder during the inference stage and does not modify the inference procedures of any baseline, thus having no impact on inference speed. For more details on the runtime analysis, please refer to Appendix B.1.

### 4.2 Main results

**Generality of SynCL.** We first evaluate SynCL on the nuScenes validation set. Tab. 1 presents the comparative results after integrating SynCL into various trackers, with each baseline shown at the top of each detector group. The experimental results lead to four key observations: **1**) SynCL can be adapted to various *tracking-by-attention* paradigm methods. Specifically, SynCL improves MUTR3D and PF-Track by $+3.7\%$ and $+3.9\%$ AMOTA, respectively. **2**) SynCL exhibits compatibility with various detectors utilizing different attention variants, indicating that the issues associated with the self-attention mechanism are widespread. Building upon our Baseline#1 and Baseline#2, SynCL achieves additional $+2.5\%$ and $2.2\%$ boosts in AMOTA. **3**) SynCL does not significantly increase the number of false positives while consistently improving the rate of recall. This indicates that our dynamic query filtering can select more high-quality candidates. **4**) As a synergistic training strategy,

Table 2: Comparison with leading camera-based methods on the nuScenes *val* set. The FPS is measured on a single NVIDIA A100 GPU, from the input images to the final tracking results. * denotes results from [25]

| Method | Backbone | Detector | Resolution | **AMOTA** | **AMOTP↓** | Recall | MOTA | IDS↓ | FP↓ | FN↓ | FPS |
|---|---|---|---|---|---|---|---|---|---|---|---|
| CC-3DT [39] | R101 | BEVFormer | 900 × 1600 | 42.9% | 1.257 | 53.4% | 38.5% | 2219 | - | - | - |
| DQTrack [11] | V2-99 | PETRv2 | 320 × 800 | 44.6% | 1.251 | - | - | 1193 | - | - | 8.6 |
| MUTR3D* [4] | V2-99 | PETR | 640 × 1600 | 44.3% | 1.299 | 55.2% | 41.6% | **175** | **11943** | 36861 | 6.1 |
| PF-Track [5] | V2-99 | PETR | 640 × 1600 | 47.9% | 1.227 | 59.0% | 43.5% | 181 | 16149 | 32778 | 5.2 |
| ADATrack++ [7] | V2-99 | PETR | 640 × 1600 | 50.4% | 1.197 | 60.8% | 44.5% | 613 | 14839 | 30616 | 3.2 |
| OneTrack [8] | V2-99 | Stream | 640 × 1600 | 54.8% | 1.088 | 61.8% | 47.9% | 389 | - | - | - |
| SynCL (ours) | V2-99 | PETR | 640 × 1600 | 50.7% | 1.183 | 61.3% | 46.2% | 248 | 14506 | 30577 | 5.2 |
| SynCL (ours) | V2-99 | Stream | 640 × 1600 | **58.9**% | **1.016** | **64.0**% | **51.5**% | 652 | 13946 | **27330** | 5.7 |

SynCL uniquely enhances both detection and tracking performance, providing a novel solution for end-to-end tracking. Notably, SynCL delivers 2% NDS improvements compared to the PF-Track.

**State-of-the-art Comparison.** In Tab. 2, we first compare SynCL with leading camera-based 3D MOT methods on the nuScenes validation set. With the PETR detector configuration, SynCL achieves the highest performance in both AMOTA and AMOTP. Remarkably, SynCL does not use the additional association module, resulting in lower inference latency compared to the current leader, ADA-Track++ [7]. With the StreamPETR detector configuration, SynCL significantly outperforms the previous state-of-the-art OneTrack by a margin of 4.1% in AMOTA.

Table 3: Comparison on nuScenes *test* set. 'E2E' stands for the end-to-end detection and tracking model.

| Method | E2E | **AMOTA** | **AMOTP↓** | Recall | MOTA |
|---|---|---|---|---|---|
| CC-3DT [39] | ✗ | 41.0% | 1.274 | 53.4% | 38.5% |
| PF-Track [5] | ✓ | 43.4% | 1.252 | 53.8% | 37.8% |
| STAR-Track [24] | ✓ | 43.9% | 1.256 | 56.2% | 40.6% |
| ADATrack++ [7] | ✓ | 50.0% | 1.144 | 59.5% | 45.6% |
| DQTrack [11] | ✓ | 52.3% | 1.096 | 62.2% | 44.4% |
| OneTrack [8] | ✓ | 55.4% | 1.021 | 60.8% | 46.1% |
| DORT [18] | ✗ | 57.6% | **0.951** | 63.4% | 48.4% |
| SynCL (ours) | ✓ | **58.8**% | 0.976 | **67.1**% | **50.4**% |

Moreover, we present comparison results on nuScenes test set in Tab. 3. SynCL achieves leading performance with 58.8% AMOTA and excels over the state-of-the-art camera-based methods. Notably, SynCL surpasses the dominant two-stage tracker [18] by 1.2% in AMOTA.

**Computation and memory complexity.** We evaluate the training costs of the models in Tab. 1. We implement the parallel decoder in an efficient way following [40]. As a result, SynCL only takes an acceptable increase in training cost including GPU memory and training time as shown in Fig. 5. For example, with Baseline#1 and MUTR3D, training time is increased by 0.3 and 0.4 hours per epoch. The memory increases are just 4G and 1G, respectively. Thus, the weight-shared decoder design slightly increases the computational cache.

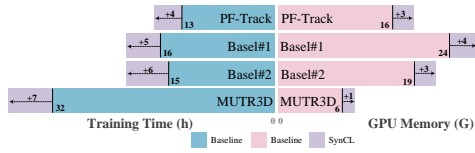

Figure 5: Analysis of training time (h) and GPU memory (G). The inference speed remains unchanged.

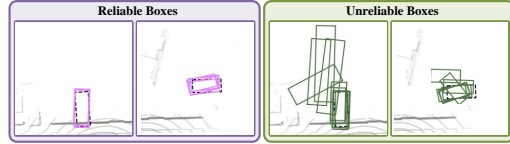

Figure 6: Illustration of our Dynamic Query Filtering with Gaussian Mixture Modeling.

### 4.3 Ablations and Analysis

**Component effectiveness.** In Tab. 4a, we analyze the importance of each proposed component in SynCL for the model performance. According to the specific task, we decompose hybrid matching into one-to-many learning for detection and one-to-one learning for tracking. In general, our proposed components enjoy consistent performance improvements. Specifically, by introducing hybrid matching, it outperforms the baseline by 1.8% AMOTA. Further integrating the instance-aware contrastive learning brings an extra 2.1% improvement in AMOTA. Together, our proposed components complement each other to effectively address multi-task learning challenges.

**Variants of query filtering.** To our knowledge, one-to-many matching has received limited attention in 3D perception. Therefore, we compare our GMM-based filtering module with methods [41–43] from 2D tasks. Concretely, we divide these methods into IoU-based group and cost-based group. As

Table 4: Ablation study on components and query filtering methods. The terms "Det" and "Trk" denote the matching for detection and tracking in Task-specific Hybrid Matching module, respectively. "CL" denotes the Instance-aware Contrastive Learning for object and track queries in C-decoder.

(a) Evaluation of component effectiveness.

| # | Det | Trk | CL | Tracking | | Detection | |
|---|---|---|---|---|---|---|---|
| | | | | AMOTA | AMOTP↓ | NDS | mAP |
| ① | | | | 40.8% | 1.343 | 47.7% | 37.8% |
| ② | ✓ | | | 42.2% | 1.327 | 49.8% | 38.9% |
| ③ | | ✓ | | 40.9% | 1.339 | 48.1% | 37.9% |
| ④ | ✓ | ✓ | | 42.6% | 1.286 | **49.9%** | **39.9%** |
| ⑤ | ✓ | ✓ | ✓ | **44.7%** | **1.262** | 49.7% | 39.6% |

(b) Comparison of query filtering methods.

| Method | Tracking | | | Detection | |
|---|---|---|---|---|---|
| | AMOTA | AMOTP↓ | Recall | NDS | mAP |
| *IoU-based* | | | | | |
| ATSS [41] | 39.3% | 1.367 | 49.9% | 47.0% | 37.3% |
| SimOTA [42] | 42.8% | 1.323 | 52.5% | 49.2% | 38.8% |
| *Cost-based* | | | | | |
| DETA [43] | 43.1% | 1.320 | 54.2% | 49.4% | 38.6% |
| GMM (ours) | **44.7%** | **1.262** | **56.5%** | **49.7%** | **39.6%** |

Table 5: Comparision results of matching variants for object queries in Task-specific Hybrid Matching module.

| # | Decoder Type | Matching Range | Strategy | Tracking | | | | | Detection | |
|---|---|---|---|---|---|---|---|---|---|---|
| | | | | AMOTA | AMOTP↓ | Recall | FP↓ | FN↓ | NDS | mAP |
| ① | S-decoder | New-born GTs | one-to-one | 40.8% | 1.343 | 50.7% | 15288 | 40398 | 47.7% | 37.8% |
| ② | S-decoder | New-born GTs | one-to-many | 38.8% | 1.357 | 52.4% | 15461 | 39930 | 46.5% | 37.1% |
| ③ | C-decoder | New-born GTs | one-to-many | 41.1% | 1.339 | 52.4% | **14265** | 40231 | 47.8% | 38.3% |
| ④ | C-decoder | All GTs | one-to-many | 42.6% | 1.300 | 53.3% | 14683 | 37666 | 49.1% | 38.6% |
| ⑤ | C-decoder | Persistent GTs | one-to-many | **44.7%** | **1.262** | **56.5%** | 15344 | **36801** | **49.7%** | **39.6%** |

Table 6: Ablation of maximum assignment constraint $K$.

| $K$ | Tracking | | | Detection | |
|---|---|---|---|---|---|
| | AMOTA | AMOTP↓ | Recall | NDS | mAP |
| 4 | 44.3% | 1.286 | 54.9% | **50.1%** | 39.5% |
| **5** | **44.7%** | **1.262** | **56.5%** | 49.7% | **39.6%** |
| 6 | 42.8% | 1.298 | 53.4% | 49.7% | 39.2% |

Table 7: Ablation of training sample length $T$.

| $T$ | Tracking | | | Detection | |
|---|---|---|---|---|---|
| | AMOTA | AMOTP↓ | Recall | NDS | mAP |
| 2 | 43.1% | 1.320 | 54.2% | 49.4% | 38.6% |
| **3** | **44.7%** | **1.262** | **56.5%** | **49.7%** | **39.6%** |
| 4 | 44.2% | 1.282 | 54.1% | 50.1% | 39.3% |

shown in Tab. 4b, IoU-based methods fail to surpass even the fixed-threshold method, DETA [43]. We infer that the object queries cannot ensure sufficient IoU overlap with GT boxes. Compared to our dynamic method, DETA proves inadequate for mining potential high-quality candidates. We further demonstrate the candidate selection effect of our GMM-based filtering module in Fig. 6.

**Variants of matching.** In Tab. 5, we analyze the impact of different matching variants for object queries in hybrid matching module. We consider from the perspectives of decoder type, matching range, and matching strategy, compared to our baseline (highlighted in gray). Based on the experimental results we draw the following two conclusions: **1**) The one-to-many assignment is not applicable to the standard decoder due to the NMS effect of the self-attention mechanism (①*vs.*②). **2**) Object queries in the C-decoder cannot benefit from a broader matching range with all ground-truth targets (④*vs.*⑤). We attribute this to the reduction of positive sample proportion in contrastive learning, which may diminish the effectiveness of synergistic learning. Moreover, matching solely with New-born GTs offers limited improvement due to the reduction in positive supervision (③*vs.*⑤).

**Hyperparameters.** We study two types of hyperparameters, the maximum assignment constraint $K$ per GT in Eq. 9 and the training sample length $T$. For $K$, according to Tab. 6, the best performance is achieved when $K = 5$. Possibly, a smaller matching number could lack supervision signals, while a larger matching number could introduce noisy and unreliable candidates. For $T$ in Tab. 7, we observe that when the frame number is increased from 2 to 3, SynCL obtains a boost of $+1.6\%$ AMOTA. However, when $T > 3$, SynCL cannot improve the model further. We infer that more consecutive frames lead to more optimization targets and temporal gradients, affecting the training stability. We thus use $K = 5$ and $T = 3$ as the default setting for all our main results and other ablation studies.

**Comparison to OneTrack.** While both OneTrack [8] and SynCL aim to optimize the joint training of detection and tracking, we summarize the core differences between SynCL and OneTrack as follows:

Table 8: Comparison of adopting SynCL with Onetrack on nuScenes *val* set. * denotes our reproduced results.

| # | Methods | Backbone | Detector | Tracking | | Detection | | #Epochs |
| | | | | AMOTA | AMOTP↓ | NDS | mAP | |
|---|---------|----------|----------|---------|---------|--------|--------|---------|
| ① | Baseline | V2-99 | PETR | 36.5% | 1.411 | 46.2% | 36.9% | 24 |
| ② | OneTrack* [8] | V2-99 | PETR | 37.8% | 1.380 | 47.3% | 37.7% | 24 |
| ③ | SynCL | V2-99 | PETR | **42.4**% | **1.347** | **48.9**% | **38.4**% | 24 |
| ④ | Baseline | V2-99 | StreamPETR | 43.8% | 1.265 | 53.6% | 45.1% | 24 |
| ⑤ | OneTrack* [8] | V2-99 | StreamPETR | 45.3% | 1.237 | 55.9% | 46.4% | 24 |
| ⑥ | SynCL | V2-99 | StreamPETR | **48.1**% | **1.204** | **56.8**% | **47.0**% | 24 |

**(1)** Research perspective. OneTrack mainly attributes the joint optimization problem to the conflicting gradient flow within the classification heads, whereas we analyze this problem from the perspective of the attention mechanism and the bipartite matching strategy.

**(2)** Orthogonality. We speculate that the multi-task learning conflict studied by both OneTrack and ourselves is orthogonal. To validate this, we reproduce the one-stage training based on PF-Track-S and our Baseline#2 (① and ④). We then reproduce OneTrack (② and ⑤) with two classification heads for gradient coordination and we do not use dynamic mask and high-cost assignment suppression. Finally, we apply our method to the reproduced OneTrack framework. As shown in Tab. 8, the consistent improvements (②*vs.*③, ⑤*vs.*⑥) prove our speculation and thus highlight our work's novelty.

**(3)** Training cost and performance. The training epochs of the one-stage training equal the total epochs of the two-stage training. Therefore, with comparable training cost, the models integrated with SynCL outperform OneTrack under the same configurations as shown in Tab. 2.

## 5  Conclusion

In this paper, we have delved into the optimization difficulties of joint detection and tracking training in end-to-end 3D trackers. We reveal that the self-attention mechanism in the transformer-based decoder hinders multi-task learning through over-deduplicating object queries and introducing self-centric attention to track queries. Based on these analyses, we have proposed to synergistically optimize multi-task learning through hybrid matching for both queries with dynamic filtering and contrastive learning within a weight-shared parallel decoder without self-attention. Both quantitative and qualitative results on the nuScenes dataset demonstrate the superiority of SynCL with consistent gains, achieving new state-of-the-art performance for the multi-camera 3D MOT task.

**Limitations and broader impacts.** The parallel decoder's construction with SynCL components leads to higher training costs, including increased training time and memory usage. Future work could investigate decoupled incremental training methods, e.g., LoRA, to further reduce these additional training expenses. Moreover, we have preliminarily validated SynCL's effectiveness under the one-stage training setting. However, the multi-task learning conflicts may not be fully resolved, as evidenced by the decreased performance compared to our two-stage training. Future work could explore more fine-grained training strategies designed for this long-term one-stage training.

## Acknowledgments and Disclosure of Funding

This work was supported in part by the Beijing Natural Science Foundation (Grant No. JQ22014, L223003, L223020), the Natural Science Foundation of China (Grant No. 62422317, U22B2056, 62036011, 62192782, U24A20331, U2441241).

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
