# OpenReview forum: "SynCL: A Synergistic Training Strategy with Instance-Aware Contrastive Learning for End-to-End Multi-Camera 3D Tracking"
_NeurIPS.cc/2025/Conference — NeurIPS 2025 poster_

### Official Review · Reviewer_A2ku · 2025-06-27

**Clarity:** 4
**Significance:** 2
**Originality:** 2
**Rating:** 5
**Confidence:** 5

**Summary:**

This paper first analyzed two major issues of the popular tracking-by-attention paradigm in 3D Multi-Object Tracking (MOT), i.e. the over duplication for object queries and the self-centric attention for track queries. Based on the investigations, the authors proposed a plug-and-play training strategy to overcome the above-mentioned issues. The proposed training strategy is targeted on the multi-task learning for detection and tracking, where a task-specific hybrid matching module is applied which uses an additional weight-shared decoder combined with an one-to-many matching for object queries for target-assignment during training. The resulting duplications are filtered using a proposed Dynamic Query Filtering module. An Instance-aware Contrastive Learning is proposed to improve the query feature discriminability. The proposed method consumes reasonable additional training cost while maintaining the same inference scheme and achieving significant performance gain.

**Questions:**

1. Line 86 the authors claimed that tracking-by-detection can be combined with query-based detectors. However, some of the following listed works [19, 20] are learned data association works, not query-based detectors.
2. Line 98, "increase inference speed" seems to be wrong. Maybe increase inference time?
3. It is unclear whether the Dynamic Query Filtering is applied only during training or also during inference.
4. In line 198, it is unclear why $$ \sum_{i=1}^2 \pi_i = 1$$.
5. What is the linear mapping function in Equ. (12) and (13)?

**Ethical Concerns:**

["NO or VERY MINOR ethics concerns only"]

**Final Justification:**

I would like to appreciate the authors to provide a detailed rebuttal and conduct comprehensive experiments, which addresses all of my concerns. The main points that I found very well in the rebuttal are:
- I believe the reproduction of OneTrack in the first table requires much effort, while a better interpretation is somehow enough to set SynCL apart from OneTrack.
- The experiment results in the second table can well support the proposed idea and enhances the interpretability of the idea.
- All the minor concerns are also very addressed.
- Concerns of other reviewers seem to be also well addressed.
I hope the authors could add the insights including experiments and discussions, as well as improvements for writing in the revised version, if the paper is accepted.

**Limitations:**

Yes.

**Paper Formatting Concerns:**

No.

**Quality:**

3

**Strengths And Weaknesses:**

**Strengths**

1. The proposed method is well-targeted, where advancing the tracking-by-attention paradigm is an interesting but less investigated research problem.
2. The paper is well presented - It first reveals and analyzes the existing problems, and then the proposed method is designed to solve the problems from the analysis, and finally validates the claims with extensive experiments.
3. While achieving state-of-the-art performance, the authors demonstrated acceptable computational overhead only during training process.
4. The paper is well written and easy to follow.

**Weaknesses**

1. The proposed parallel decoder and hybrid matching design shares some similarities to OneTrack [8], which uses two classification heads and a matching with gradient coordination. The purpose of the proposed components in this work also aligns with OneTrack, where OneTrack allows a one-stage end-to-end training, i.e. the pre-training of object detection is unnecessary. It is worth to conduct a more in-depth comparison to OneTrack from both methodological and experimental perspectives. Without such convincing comparison, the novelty of this paper will be limited.
2. The authors claimed that the over-deduplication of object queries and the self-centric attention of track queries are caused by the inconspicuous nature of the self-attention mechanism (line 147, line 13). However, this conclusion is not precise enough since the self-attention combined with the bipartite matching together cased the problem, not the self-attention itself. In fact, self-attention allow information exchange between queries which could benefit the learning process. Therefore, I have the doubts about whether the self-attention layer in the hybrid matching module should be removed. Also, the claims of "self-attention is the main cause of the problems in tracking-by-attention" should be rephrased in the whole paper.
3. The proposed Instance-aware Contrastive Learning seems like a plug-and-play component in parallel to the main contributions. It could also benefit other MOT methods and this should be validated by experiments.

---

> ### Author Rebuttal · Authors · 2025-07-31
>
> Thank you for your insightful review and valuable comments! We are pleased that you acknowledge our paper's *good* quality, *excellent* clarity, and contribution in *advancing the tracking-by-attention paradigm* from an *interesting and less investigated research perspective*. We will respond to your concerns point by point.
>
> > Q1: It is worth to conduct a more in-depth comparison to OneTrack from both methodological and experimental perspectives.
>
> * We appreciate your highlighting of this concern. While both OneTrack and our proposed SynCL aim to optimize the joint training of detection and tracking, we summarize the core differences between SynCL and OneTrack as follows:
>     * **Research perspective.** OneTrack mainly attributes the joint optimization problem to the conflicting gradient flow within the classification heads, whereas we analyze this problem from the perspective of the attention mechanism and the bipartite matching strategy.
>     * **Orthogonality.** We speculate that the multi-task learning conflict studied by both OneTrack and ourselves is orthogonal. However, **OneTrack has not yet released its open-source code**. To validate this, we reproduce the one-stage training (① and ④) based on PF-Track-S and our Baseline#2. We then reproduce OneTrack (② and ⑤) with two classification heads for gradient coordination as one of the core contributions in their paper and we do not use *dynamic mask* and *high-cost assignment suppression* due to limited rebuttal time. Finally, we apply our method to the reproduced OneTrack framework (③ and ⑥). As shown in the table below, the consistent improvements (② *vs.* ③, ⑤ *vs.* ⑥) prove our speculation and thus highlight our work's novelty.
>     * **Generalization.** The solution in OneTrack has only been validated within the StreamPETR detector, whereas SynCL can be integrated into any tracking-by-attention paradigm tracker and has been validated within multiple detectors. This has also been acknowledged by Reviewer `FdMH` and Reviewer `rfd7`.
>     * **Training cost and performance.** The training epochs of the one-stage training equal the total epochs of the two-stage training. Therefore, with comparable training cost, the models integrated with SynCL outperform OneTrack under the same detector and resolution configurations as shown in Table 1 and 2 of our main paper.
>
>     | # | Methods | Backbone | Detector | AMOTA | AMOTP↓ | NDS | mAP | #Epochs |
>     | - | - | - | - | - | - | - | - | - |
>     | ① | Baseline | V2-99 | PETR | 36.5% | 1.411 | 46.2% | 36.9% | 24 |
>     | ② | OneTrack (reproduced) | V2-99 | PETR | 37.8% | 1.380 | 47.3% | 37.7% | 24 |
>     | ③ | SynCL | V2-99 | PETR | **42.4**% | **1.347** | **48.9**% | **38.4**% | 24 |
>     | ④ | Baseline | V2-99 | StreamPETR | 43.8% | 1.265 | 53.6% | 45.1% | 24 |
>     | ⑤ | OneTrack (reproduced) | V2-99 | StreamPETR | 45.3% | 1.237 | 55.9% | 46.4% | 24 |
>     | ⑥ | SynCL | V2-99 | StreamPETR | **48.1**% | **1.204** | **56.8**% | **47.0**% | 24 |
>
> * We will follow your advice to include these more in-depth and convincing analyses in the final revision.
>
> > Q2: I have the doubts about whether the self-attention layer in the hybrid matching module should be removed. Also, the claims should be rephrased in the whole paper.
>
> * We appreciate your constructive advice and agree with your key point. We also believe that the bipartite matching determines the optimization objective, which in turn leads to the nature of the self-attention mechanism. However, we find that the self-attention mechanism and bipartite matching are **tightly coupled in the standard decoder**. Directly modifying the matching strategy without changing the decoder architecture is not feasible. We demonstrate the findings with the following key experiments:
>
>     * As shown in row 2 of Table 5 in our main paper, we have applied one-to-many matching to object queries in the standard decoder (S-decoder) **without constructing the parallel decoder**, but this results in a decrease in model performance (① *vs.* ②).
>     * We further try to expand the matching range of object queries to include all GTs while still keeping the self-attention mechanism in the only standard decoder. However, both the detection and tracking performance of the model suffered significantly (② *vs.* ③). This shows that allowing object queries and track queries to match the same ground truth directly in the standard decoder worsens the query competition rather than improving it.
>
>     | # | Decoder Type | Matching Range | Strategy | AMOTA | AMOTP↓ | NDS | mAP |
>     | - | - | - | - | - | - | - | - |
>     | ① | S-decoder | New-born GTs | one-to-one | **40.8**% | **1.343** | **47.7**% | **37.8**% |
>     | ② | S-decoder | New-born GTs | one-to-many | 38.8% | 1.357 | 46.5% | 37.1% |
>     | ③ | S-decoder | All GTs | one-to-many | 26.7% | 1.470 | 38.5% | 17.3% |
>
>     * Finally, we retain the self-attention mechanism in the parallel decoder. Compared to the baseline (④ *vs.* ⑤), this also leads to a severe decrease in model performance, demonstrating that the query competition issues remain with the existence of the self-attention mechanism.
>
>     | # | Parallel Decoder Attention Mechanisim | AMOTA | AMOTP↓ | NDS | mAP |
>     | - | - | - | - | - | - |
>     | ④ | cross | **44.7**% | **1.262** | **49.7**% | **39.6**% |
>     | ⑤ | self+cross | 28.1% | 1.458 | 39.9% | 17.8% |
>
> * These experimental results indicate that it is challenging to address the problems caused by the self-attention mechanism and bipartite matching together without removing the self-attention mechanism. We also agree that the self-attention mechanism can benefit the learning process apart from the over-deduplication and self-centric issues. Based on these observations, we propose SynCL **in a decoupled manner**:
>
>     * We have **retained the self-attention mechanism** in the standard decoder to continue leveraging its advantages.
>     * We further introduce a parallel decoder combined with a hybrid matching strategy to indirectly but effectively mitigate the identified issues (as illustrated in Figure 1 and 4). This rational designation has also been acknowledged by Reviewer `FdMH`.
>
> * Following your advice, we will rephrase all the misleading claims with regard to "self-attention is the main cause" and emphasize the effect of bipartite matching (e.g., line 13, line 147) in the revision.
>
> >Q3: Instance-aware Contrastive Learning seems like a plug-and-play component in parallel to the main contributions. It could also benefit other MOT methods and this should be validated by experiments.
>
> * Thank you for your suggestions. We would like to clarify that our proposed Instance-aware Contrastive Learning is **not an independent component** from the following aspects.
>     * Firstly, the partition of positive and negative sample pairs in Instance-aware Contrastive Learning relies on our proposed hybrid matching strategy, which ensures that a ground-truth can be simultaneously assigned to both an object query and a track query.
>     * Secondly, without our Dynamic Query Filtering to identify high-quality candidates, Instance-aware Contrastive Learning cannot effectively facilitate information transfer between track queries and object queries. As shown in Table 4(b) of our main paper, other variants of query filtering are significantly less effective than our GMM-based method.
>     * These interdependent components highlights the coherent nature of our designation. Together, these three components of SynCL can serve as a plug-and-play training strategy that optimizes any tracking-by-attention paradigm method, as shown in Table 1 of our main paper.
>
> * Following your advice, we have successfully integrated SynCL into an additional MOT method, i.e., OneTrack (**in Q1**). These experimental results demonstrate the generality and novelty of our work.
>
> >Q4: Line 86, some of the following listed works [19, 20] are learned data association works, not query-based detectors. Line 98, "increase inference speed" seems to be wrong.
>
> We are sorry for the confusion caused in lines 86 and 98 and we will revise them as follows:
> * Line 86: We will rephrase it to "*Tracking-by-detection* paradigm achieves further advancements when combined with learning-based association methods."
> * Line 98: Thank you for your feedback. The correct statement should be "increase inference time".
>
> > Q5: It is unclear whether the Dynamic Query Filtering is applied only during training or also during inference.
>
> We appreciate your attention to this concern. Dynamic Query Filtering is applied only during training based on the cost matrix between ground-truths and predictions, without affecting the inference process. Following your advice, we will emphasize this in the revision.
>
> > Q6: About the weight normalization in line 198.
>
> We appreciate your rigorous consideration. In probability theory, a valid probability distribution function must integrate to 1. The weights in the mixture model represent the relative importance that each Gaussian component contributes to the entire model. Therefore, their total must equal 1 to ensure that the mixture model is a legitimate probability distribution.
>
> > Q7: What is the linear mapping function in Equ. (12) and (13)?
>
> Thank you for pointing this out. The linear mapping function is described in lines 215-216 of our main paper and we will make this more clear in the revision.

---

> > ### Comment · Reviewer_A2ku · 2025-08-07
> >
> > I would like to appreciate the authors for their efforts in providing an excellent rebuttal. As my concerns are well addressed (see final justifications), I will raise my rating to Accept.

---

> > > ### Author Response · Authors · 2025-08-07
> > > **Thank Reviewer A2ku for raising the rating to Accept**
> > >
> > > We are glad that our clarifications addressed your concerns well and grateful for your increased score. In the final version, we will incorporate the additional results and technical details following your suggestions.

---

> ### Author Response · Authors · 2025-08-05
>
> Dear Reviewer `A2ku`,
>
> I hope this message finds you well. As the discussion period is nearing its end, we would like to ensure we have addressed all your concerns satisfactorily.
>
> We deeply appreciate the time and thought you have dedicated to reviewing our paper. If there are any additional points or feedback you'd like us to consider, please let us know. Your insights are invaluable to us, and we're eager to address any remaining issues to improve our work.
>
> Thank you again for your valuable feedback and support throughout this process.
>
> Best regards,
>
> The authors of Paper 4526

---

### Official Review · Reviewer_VTpN · 2025-07-01

**Clarity:** 2
**Significance:** 2
**Originality:** 3
**Rating:** 4
**Confidence:** 3

**Summary:**

This paper points out that self-attention causes issues like over-deduplication for object queries and self-centric attention for track queries. To solve this, this paper introduces SynCL, a new training strategy that improves the interaction between detection and tracking. The approach includes a task-specific hybrid matching module, a dynamic query filtering module, and an instance-aware contrastive learning method. These methods help enhance the representation learning for both detection and tracking tasks by refining candidate selection, reducing redundant predictions, and improving the cross-task connections.

**Questions:**

Please refer to the Weaknesses section for my main concerns.
Additionally, as I am not deeply familiar with the 3D tracking field, I may take into account the opinions of other reviewers when adjusting the final score.

**Ethical Concerns:**

["NO or VERY MINOR ethics concerns only"]

**Final Justification:**

After considering the authors' rebuttal and the discussion with other reviewers, I maintain my original weak accept rating.

**Limitations:**

yes

**Paper Formatting Concerns:**

No paper formatting concerns.

**Quality:**

3

**Strengths And Weaknesses:**

Strengths:
1.	The paper presents an interesting observation regarding how self-attention contributes to the optimization difficulty in joint detection and tracking tasks.
2.	The proposed method achieves good performance.
3.	The paper is well-written, and the experiments are thorough and comprehensive.

Weaknesses:
Although the paper offers a valuable insight into the limitations of self-attention, the proposed solution mainly involves removing the self-attention module. Other components (Task-specific Hybrid Matching, Dynamic Query Filtering, and Instance-aware Contrastive Learning) appear to be loosely related to the identified problem.

---

> ### Author Rebuttal · Authors · 2025-07-31
>
> Thank you for your insightful review and valuable comments! We are pleased that you find our observation in joint detection and tracking learning *interesting*. We also value your appreciation of our *thorough and comprehensive experiments* and *good performance*. We will respond to your concerns point by point.
>
> > Q1: The proposed solution mainly involves removing the self-attention module. Other components appear to be loosely related to the identified problem.
>
> * We are sorry for not highlighting the relationships among the components in SynCL. Firstly, we would like to emphasize that our core findings are not limited to the self-attention mechanism as follows:
>     * The self-attention mechanism serves as our **entry point** for exploring the challenges of multi-task learning within the tracking-by-attention paradigm. From this perspective, we analyze the role of different attention mechanisms under the original bipartite matching strategy in facilitating end-to-end detection and tracking.
>     * We find that while the self-attention mechanism coordinates the functions of different types of queries, it also introduces issues, e.g., the over-deduplication of object queries and self-centric attention of track queries.
>     * These issues stem directly from the self-attention mechanism and the bipartite matching strategy. However, the deeper underlying causes are inherent challenges within multi-camera 3D tracking, e.g., a lack of 3D spatial geometric knowledge, depth estimation capability and difficulties in associating the same target across multiple views and frames.
>     * Our proposed SynCL provides a comprehensive and synergistic method to address not only the limitations of the self-attention mechanism but also these broader challenges.
>
> * Secondly, we are more than willing to further clarify the role of each component within SynCL and the specific problems they aim to resolve:
>     * **Task-specific Hybrid Matching.** To recover over-deduplicated object queries, we enhance the supervision of object query predictions through a one-to-many matching strategy in the parallel decoder. Simultaneously, to improve the model's inter-temporal association capability, we employ identity-guided one-to-one matching for track queries.
>     * **Dynamic Query Filtering.** To uncover more high-quality candidates with enhanced 3D geometric knowledge and depth-aware information, we propose a GMM-based filtering method, dynamically selecting reliable 3D predictions for the one-to-many matching, as shown in Figure 6 of our main paper and Appendix Figure A1.
>     * **Instance-aware Contrastive Learning.** To break the constraints of self-centric attention on track queries and improve inter-frame and cross-view association capability, SynCL introduces a novel Instance-aware Contrastive Learning component. This contrastive learning aligns track queries with the high-quality 3D object query features, addressing the issue of low-quality predictions for track queries caused by occlusions or motion blur during temporal propagation.
>
> * Lastly, we summarize the relationships and underlying design logic among the components of SynCL as follows:
>     * The tight coupling between the self-attention mechanism and bipartite matching within the standard decoder makes it impractical to directly apply our Task-specific Hybrid Matching without the parallel decoder, as shown in the table in **response to Q2 of Reviewer `A2ku`**.
>     * The Task-specific Hybrid Matching in the parallel decoder ensures that the matching ranges of object queries and track queries have an intersection, enabling the partitioning of positive and negative sample pairs in Instance-aware Contrastive Learning.
>     * The benefits brought by Instance-aware Contrastive Learning are mainly determined by the effectiveness of query filtering. The higher the quality of the selected object queries, the more correction information can be obtained for track queries. As shown in Table 4(b) of our main paper, other query filtering methods are significantly less effective in comparison to our GMM-based method.
>
> * These well-targeted and interdependent components highlight the coherent nature of our designation, which has also been acknowledged by Reviewer `FdMH`. We will emphasize the goals of each component within SynCL and the inner relationships among them in the revision. We will also swap the order of Section 3.1 (Analysis) and Section 3.2 (Preliminaries) to first establish the foundational knowledge of the tracking-by-attention paradigm before analyzing and introducing our work's motivation, following the advice of Reviewer `FdMH`.

---

> ### Author Response · Authors · 2025-08-05
>
> Dear Reviewer `VTpN`,
>
> I hope this message finds you well. As the discussion period is nearing its end, we would like to ensure we have addressed all your concerns satisfactorily.
>
> We deeply appreciate the time and thought you have dedicated to reviewing our paper. If there are any additional points or feedback you'd like us to consider, please let us know. Your insights are invaluable to us, and we're eager to address any remaining issues to improve our work.
>
> Thank you again for your valuable feedback and support throughout this process.
>
> Best regards,
>
> The authors of Paper 4526

---

> ### Comment · Area_Chair_GS2D · 2025-08-07
>
> Hi Reviewer VTpN,
>
> Please check the author's feedback, evaluate how it addresses the concerns you raised, and discuss the rebuttal with the authors. Please do this ASAP.
> Thanks.
>
> Your AC.

---

> ### Comment · Reviewer_VTpN · 2025-08-07
>
> Thank you for the rebuttal. I have no further questions and will keep my weak accept rating.

---

> > ### Author Response · Authors · 2025-08-07
> > **Thank Reviewer VTpN for the positive recommendation**
> >
> > Thank you for maintaining your initial weak accept rating and increasing the Confidence score from 2 to 3 after checking our feedback. We will incorporate the additional results and technical details following your suggestions.

---

### Official Review · Reviewer_FdMH · 2025-07-02

**Clarity:** 4
**Significance:** 4
**Originality:** 4
**Rating:** 4
**Confidence:** 5

**Summary:**

This paper handles the over-deduplication for object queries and self-centric attention for track queries caused by the self-attention mechanism for robust multi-view 3D object tracking. The authors present SynCL, a novel plug-and-play synergistic training strategy designed to co-facilitate multi-task learning for detection and tracking. A Task-specific Hybrid Matching module is designed to exploit promising candidates overlooked by the self-attention mechanism, as well as a Dynamic Query Filtering module to select optimal candidates for the one-to-many matching. Besides, the Instance-aware Contrastive Learning is applied to bridge the gap between detection and tracking. Following the tracking-by-attention paradigm, experimental results on nuScenes demonstrate the effectiveness of the proposed method to improve the baselines with the state-of-the-art performances.

**Questions:**

I do have some questions, see those questions in the weakness part.

**Ethical Concerns:**

["NO or VERY MINOR ethics concerns only"]

**Final Justification:**

After considering the authors' rebuttal and the discussion with other reviewers, I maintain my initial Borderline accept rating.

**Limitations:**

I don’t see the discussion related to the limitations and broader impact. I hope the authors can provide a brief discussion on this part in the response and are encouraged to detail this part in the revision.

**Quality:**

3

**Strengths And Weaknesses:**

*Strengths:
1. Enough Novelty: Observing the impedance of the self-attention mechanism across different queries to joint detection and tracking training, this paper proposes a synergistic training strategy compatible with any tracking-by-attention paradigm tracker to address multi-task learning challenges.
2. Rational Designation: The proposed SynCL implements dynamic filtering-based hybrid matching and instance-aware contrastive learning to recover over-deduplicated object queries and improve the classification capability of matching object with tracking queries, which is intuitively beneficial for tracking.
3. Rich Experiments and Analysis. The authors provide extensive experiments and analysis for the proposed method. I appreciate this. The experimental analysis with various ablation studies allows a better understanding of each module and overall performance.
4. Good writing and organization. This paper is well-written and organized. Each section has a clear motivation. It’s easy to follow the ideas.

*Weaknesses:
1. Imbalanced Performance Metrics: While achieving considerable performance gains in the main metrics (e.g., AMOTA and AMOTP), the IDS degrades compared with the baselines. What are the possible reasons?
2. Insufficient Preliminaries: The authors should first express the self-attention mechanism in the tracking-by-attention paradigm (better in a pipeline visualization), instead of directly casting the motivation that may distort the authors.
3. More Comprehensive Ablation: It’s better to show the generality of the proposed method in the tracking-by-detection paradigm, e.g., applying the Dynamic Query Filtering to the matching process.

---

> ### Author Rebuttal · Authors · 2025-07-31
>
> Thank you for your insightful review and valuable comments! We are happy that you acknowledge the *novelty* and *rational designation* of our paper in addressing multi-task learning challenges within the tracking-by-attention paradigm. We also value your appreciation of our *rich experiments and analysis*. We will respond to your concerns point by point.
>
> > Q1: The IDS degrades compared with the baselines. What are the possible reasons?
>
> * We appreciate your highlighting of this concern. We would like to first summarize and analyze the changes in ID switches before and after integrating our SynCL as follows:
>     * In Table 1 of our main paper, PF-Track and Baseline#1 show fewer ID switches compared to MUTR3D and Baseline#2 because the PF-Track framework includes a trajectory prediction module, which prolongs the lifespan of tracklets when occlusions or motion blur result in low-confidence predictions for track queries.
>     * With this trajectory prediction module, if the detector performs better, integrating our SynCL can actually reduce ID switches (Baseline#1 *vs.* SynCL: IDS 173 *vs.* IDS 170) and significantly improve overall metrics, including AMOTA which offers a holistic evaluation of tracking performance by accounting for IDS, FP, FN, *etc.*.
>     * Conversely, without this trajectory prediction module, even when using a stronger detector (MUTR3D *vs.* Baseline#2), integrating SynCL will still lead to an increase in ID switches.
>
> * We summarize the possible reasons for this degradation of the IDS metric as follows:
>
>     * Firstly, our Hybrid Matching module has uncovered more high-quality candidates, leading to an **increase in the model's total positive predictions**, as shown in the table below. From a probabilistic perspective, this has heightened the likelihood of ID switches occurring between tracklets.
>
>     | Methods | Backbone | Detector | IDS↓ | TP | FP↓ | Total Positive Predictions |
>     | - | - | - | - | - | - | - |
>     | MUTR3D | R101 | DETR3D | **474** | 57595 | 15269 | 72864 |
>     | SynCL | R101 | DETR3D | 588 | **60569** | **14311** | **74880** |
>     | PF-Track-S | V2-99 | PETR | **166** | 60879 | 16331 | 77210 |
>     | SynCL | V2-99 | PETR | 203 | **64893** | **15344** | **80237** |
>     | Baseline#1 | V2-99 | PETRv2 | 173 | 64659 | 14106 | 78765 |
>     | SynCL | V2-99 | PETRv2 | **170** | **65923** | **13411** | **79334** |
>     | Baseline#2 | V2-99 | Stream | **411** | 66257 | 13962 | 80219 |
>     | SynCL | V2-99 | Stream | 540 | **67689** | **13639** | **81328** |
>
>     * Secondly, track queries may generate low-quality predictions during the inter-frame propagation due to factors, e.g., occlusion and motion blur. With the integration of SynCL, object queries can discover and produce high-quality predictions for the same target while suppressing the confidence of the corresponding track query predictions, creating new tracklets. Although this substitution may lead to ID switches, it allows for **robust re-detection**. After incorporating PF-Track’s trajectory prediction module, the model can retain track queries for a longer duration before re-detecting them, facilitating the identification of truly lost targets outside the field of view. Therefore, the combination of robust re-detection with the trajectory prediction module is essential for addressing the issue of ID switches.
>
> * We will consider incorporating a similar module in our future work to further enhance the IDS performance of other MOT methods and will include these new analyses in the revision.
>
> > Q2: The authors should first express the self-attention mechanism in the tracking-by-attention paradigm (better in a pipeline visualization), instead of directly casting the motivation that may distort the authors.
>
> Thank you for your suggestions. To address this concern and enhance the paper's readability, we will make the following adjustments in the revision:
>
> * We will modify the pipeline visualization in Figure 1 to better illustrate how the self-attention mechanism operates within both the standard decoder and the parallel decoder, along with the corresponding algorithmic workflow.
>
> * We will swap the order of Section 3.1 (Analysis) and Section 3.2 (Preliminaries) to first establish the foundational knowledge of the tracking-by-attention paradigm before analyzing and introducing our work's motivation.
>
> > Q3: It’s better to show the generality of the proposed method in the tracking-by-detection paradigm, e.g., applying the Dynamic Query Filtering to the matching process.
>
> We appreciate your constructive advice. We choose the state-of-the-art tracking-by-detection paradigm method, DQTrack, as our baseline. We apply Dynamic Query Filtering to the affinity matrix in the association module to dynamically adjust the matching threshold in inference, as shown in the table below. The consistent improvements demonstrate the effectiveness and generality of our GMM-based query filtering module. We will add this new ablation comparison in the revision and include more tracking-by-detection paradigm methods in future work.
>
> | Methods | Backbone | Detector | AMOTA | AMOTP↓ | Recall | MOTA | IDS↓ |
> | - | - | - | - | - | - | - | - |
> | DQTrack | R101 | UVTR-C | 39.6% | 1.310 | 48.8% | 32.9% | 1267 |
> | +Dynamic Query Filtering | R101 | UVTR-C | **40.0**% | **1.307** | **49.1**% | **33.4**% | **1219** |
> | DQTrack | V2-99 | PETRv2 | 44.6% | 1.251 | **54.7**% | 38.1% | 1193 |
> | +Dynamic Query Filtering | V2-99 | PETRv2 | **45.1**% | **1.247** | 54.4% | **38.5**% | **1128** |
>
> > Q4: I don’t see the discussion related to the limitations and broader impact.
>
> Although we have analyzed the computation and memory complexity in Section 4.2 as the limitations we mentioned in checklist, demonstrating that our method incurs additional training costs, we summarize the limitations and broader impact of SynCL in detail as follows:
>
> * The construction of the parallel decoder combined with SynCL's components incurs additional training costs, including increased training time and memory usage. Future work could consider exploring decoupled incremental training methods, e.g., LoRA, to further reduce these additional training expenses.
> * In response to Q1 of Reviewer `A2ku`, we have preliminarily validated SynCL's effectiveness under the one-stage training setting. However, the multi-task learning conflicts may not be fully resolved, as evidenced by the decreased performance compared to our two-stage training. Future work could explore more fine-grained training strategies designed for this long-term one-stage training.
>
> Following your advice, we will include these discussions in the revision.

---

> ### Author Response · Authors · 2025-08-05
>
> Dear Reviewer `FdMH`,
>
> I hope this message finds you well. As the discussion period is nearing its end, we would like to ensure we have addressed all your concerns satisfactorily.
>
> We deeply appreciate the time and thought you have dedicated to reviewing our paper. If there are any additional points or feedback you'd like us to consider, please let us know. Your insights are invaluable to us, and we're eager to address any remaining issues to improve our work.
>
> Thank you again for your valuable feedback and support throughout this process.
>
> Best regards,
>
> The authors of Paper 4526

---

### Official Review · Reviewer_rfd7 · 2025-07-06

**Clarity:** 3
**Significance:** 2
**Originality:** 2
**Rating:** 4
**Confidence:** 3

**Summary:**

This paper points out two issues with the self-attention mechanism in tracking-by-detection model: over-deduplication for object queries and self-centric attention for existing track queries. To resolve this, a parallel weight-shared decoder that operates without self-attention is introduced during training, and a series of training strategies are applied, including Task-specific Hybrid Matching, Dynamic Query Filtering and Instance-aware Contrastive Learning. Experiments on the nuScenes dataset show that the proposed method improves performance on multiple baseline trackers.

**Questions:**

Implementation details about dynamic query filtering are missing. For example
- In cost-based filtering, what is the cost function between prediction and ground-truth ? what is the threshold to distinguish reliable and unreliable clusters ?
- In IoU-based filtering, is it normal IoU between 3D boxes or other variants, like gIoU?

**Ethical Concerns:**

["NO or VERY MINOR ethics concerns only"]

**Final Justification:**

Thank authors for the detailed response. My concerns about technical details have been addressed. Although I still think adapting 2D technology to 3D is not novel enough, I will increase the score given the good performance number.

**Limitations:**

The authors didn't discuss the limitations.

**Quality:**

3

**Strengths And Weaknesses:**

Strengths:
- This paper provides a very clear analysis of how existing tracking-by-attention models suffer from over-deduplication and self-centric attention issues. Aiming at solving it, a parallel decoder without self-attention is used during training in task-specific hybrid matching module, along with dynamic query filtering module to select reliable candidates, and instance-aware contrastive loss to force interaction between object and track queries corresponding to the same instance.

- Comprehensive experimental results are provided. The proposed method is successfully applied to various baseline models and brings convincing benefits, for example, MUTR3D, PF-Track, PETRv2, StreamPETR, demonstrating its generalization and effectiveness. Finally, 58.9% AMOTA on the nuScenes validation set and 58.8% on the test set are achieved.


Weaknesses:
- The major concern about this paper is limited novelty. Although the analysis of self-attention mechanism and a parallel decoder without self-attention during training are proved very useful, this is not the first time I see this technology. For example, DAC-DETR[1] is published two years ago.
- Most discussions in this paper are well-studied in 2D object detection and tracking filed[1][2][3][4]. Adapting these technologies from 2D to 3D seems like an incremental work, so my initial rating is borderline reject. But I will re-evaluate my rating after discussing with other reviewers.

[1] DAC-DETR: Divide the Attention Layers and Conquer

[2] MOTR: End-to-End Multiple-Object Tracking with Transformer

[3] DETA: NMS Strikes Back

[4] YOLOX: Exceeding YOLO Series in 2021:

---

> ### Author Rebuttal · Authors · 2025-07-31
>
> Thank you for your insightful review and valuable comments! We are pleased that you acknowledge our paper's *clear analysis* for existing tracking-by-attention models and *convincing benefits* across various baseline models, demonstrating the *generalization* and *effectiveness* of our method. We will respond to your concerns point by point.
>
> > Q1: The major concern about this paper's novelty is that DAC-DETR for 2D object detection also investigates the attention mechanisms and most discussions in this paper are well-studied in 2D object detection and tracking field.
>
> We are sorry for not highlighting our novelty in comparison to the previous 2D detection and tracking technologies thoroughly in our main paper. We propose SynCL as the first novel synergistic training strategy compatible with any tracking-by-attention paradigm tracker, achieving a new state-of-the-art performance of 58.9% AMOTA on the nuScenes dataset. Our novelty and advancements in the tracking-by-attention paradigm have been acknowledged by Reviewer `FdMH` and Reviewer `A2ku`. We are more than willing to further clarify our novelty and contributions as follows:
>
> * Although DAC-DETR also draws on the investigation of the attention mechanisms, we summarize the core differences between SynCL and DAC-DETR.
>     * The multi-camera 3D tracking task, addressed by SynCL, constitutes the significant **expansion in research scope and complexity** compared to 2D object detection, inherently requiring processing and integrating information across additional dimensions, e.g., temporal cues, multi-view inputs, and 3D spatial perception.
>     * We have reached **more comprehensive conclusions** specific to the 3D MOT task through our qualitative (prediction visualization in Figure 2) and quantitative (self-attention heatmap in Figure 3) analyses, e.g., the over-deduplication of object queries and the self-centric issues of track queries.
>     * As we claim in line 55-58 that DAC-DETR cannot address the issues we have identified simultaneously, we supplementarily validate this by directly applying DAC-DETR’s parallel decoder technology to the tracking-by-attention paradigm as shown in ③ of the table below.
>     * Compared to the static matching strategy (DETA) in DAC-DETR, the Dynamic Query Filtering in SynCL uncovers more high-quality candidates with enhanced 3D geometric knowledge and depth-aware information, as shown in Figure 6 of our main paper and Appendix Figure A1.
>     * SynCL further introduces a novel Instance-aware Contrastive Learning to align these high-quality 3D query features with track queries, effectively breaking the constraints of self-centric attention. Importantly, this contrastive learning is built upon our overall designation rather than an independent component, please refer to the **response to Q3 of Reviewer `A2ku` and Q1 of Reviewer `VTpN`**.
>
> * To further substantiate our contributions, we compare SynCL with other 2D object detection and tracking works [2,3,4].
>     * MOTR [2] is the first work to propose the tracking-by-attention paradigm in 2D MOT, but it does not systematically explore the challenges of joint detection and tracking training.
>     * Although DETA [3] and YOLOX [4] have designed one-to-many matching strategies (DETA and SimOTA) specifically for 2D detectors, adapting these technologies directly from 2D to 3D MOT yields limited improvements, as shown in ② and ③ of the table below.
>     * We would like to emphasize that the ablation results in Table 4(b) of our main paper **incorporate both Hybrid Matching and Instance-aware Contrastive Learning**. We combine these results below (④ and ⑤) for a thorough comparison.
>     * In contrast to the incremental adaptation of these 2D techniques to 3D MOT (② *vs.* ④, ③ *vs.* ⑤), the consistent improvements by SynCL highlight the novelty of our work and its contribution to 3D MOT.
>
>     | # | Methods | Backbone | Detector | AMOTA | AMOTP↓ | NDS | mAP |
>     | - | - | - | - | - | - | - | - |
>     | ① | PF-Track-S | V2-99 | PETR | 40.8% | 1.343 | 47.7% | 37.8% |
>     | ② | DAC-DETR (SimOTA) | V2-99 | PETR | 40.7% | 1.345 | 48.0% | 37.8% |
>     | ③ | DAC-DETR (DETA) | V2-99 | PETR | 41.3% | 1.337 | 48.7% | 38.3% |
>     | ④ | SynCL (SimOTA) | V2-99 | PETR | 42.8% | 1.323 | 49.2% | 38.8% |
>     | ⑤ | SynCL (DETA) | V2-99 | PETR | 43.1% | 1.320 | 49.4% | 38.6% |
>     | ⑥ | SynCL (GMM) | V2-99 | PETR | **44.7**% | **1.262** | **49.7**% | **39.6**% |
>
> * We have additionally integrated SynCL into the one-stage training method, i.e., OneTrack, to further demonstrate the generality and novelty of SynCL. Please refer to the Table in **response to Q1 of Reviewer `A2ku`**.
>
> * We will include these discussions in the revision (e.g., in lines 55-58 and Table 4) to highlight our novelty and contributions compared to previous 2D works.
>
> > Q2: In cost-based filtering, what is the cost function between prediction and ground-truth? what is the threshold to distinguish reliable and unreliable clusters?
>
> Thank you for pointing this out. In cost-based filtering, the cost function combines focal loss for classification and L1 loss for 3D box regression, consistent with the cost metrics in Eq. (8). As shown in the yellow box of Figure 4 in our main paper, the filtering threshold in SynCL is adaptively calculated using our GMM-based query filtering method, based on the cost matrix.
>
> > Q3: In IoU-based filtering, is it normal IoU between 3D boxes or other variants, like gIoU?
>
> Thank you for pointing this out. As shown in Table 4 (b) of our main paper, we have only replaced the query filtering component in SynCL with other variants to validate the effectiveness of our GMM-based method. All IoU-based filtering variants in Table 4 (b) utilize normal IoU to compute the IoU between prediction 3D boxes and ground-truth 3D boxes, which determines the allocation of the top-k object queries for each ground-truth. Besides, we find that other IoU types, e.g., gIoU, have a **minimal impact** on model performance, as shown below.
>
> | Query Filtering | IoU Type | AMOTA | AMOTP↓ | NDS | mAP |
> | - | - | - | - | - | - |
> | SimOTA | Normal IoU | 42.8% | 1.323 | 49.2% | 38.8% |
> | SimOTA | gIoU | 43.0% | 1.318 | 49.5% | 38.9% |
>
> > Q4: The authors didn't discuss the limitations.
>
> Although we have analyzed the computation and memory complexity in Section 4.2 as the limitations we mentioned in checklist, demonstrating that our method incurs additional training costs, we summarize the limitations and broader impact of SynCL in detail as follows:
>
> * The construction of the parallel decoder combined with SynCL's components incurs additional training costs, including increased training time and memory usage. Future work could consider exploring decoupled incremental training methods, e.g., LoRA, to further reduce these additional training expenses.
> * In response to Q1 of Reviewer `A2ku`, we have preliminarily validated SynCL's effectiveness under the one-stage training setting. However, the multi-task learning conflicts may not be fully resolved, as evidenced by the decreased performance compared to our two-stage training. Future work could explore more fine-grained training strategies designed for this long-term one-stage training.
>
> Following your advice, we will include these discussions in the revision.
>
> **References**:
>
> [1] Hu Z, Sun Y, Wang J, et al. DAC-DETR: Divide the Attention Layers and Conquer. NeurIPS, 2023.
>
> [2] Zeng F, Dong B, Zhang Y, et al. MOTR: End-to-End Multiple-Object Tracking with Transformer. ECCV, 2022.
>
> [3] Jeffrey Ouyang-Zhang, Jang Hyun Cho, Xingyi Zhou and Philipp Krähenbühl. DETA: NMS Strikes Back. arXiv: 2212.06137 (2022).
>
> [4] Ge Z, Liu S, Wang F, Li Z, and Sun J. YOLOX: Exceeding YOLO Series in 2021. arXiv:2107.08430 (2021).

---

> ### Author Response · Authors · 2025-08-05
>
> Dear Reviewer `rfd7`,
>
> I hope this message finds you well. As the discussion period is nearing its end, we would like to ensure we have addressed all your concerns satisfactorily.
>
> We deeply appreciate the time and thought you have dedicated to reviewing our paper. If there are any additional points or feedback you'd like us to consider, please let us know. Your insights are invaluable to us, and we're eager to address any remaining issues to improve our work.
>
> Thank you again for your valuable feedback and support throughout this process.
>
> Best regards,
>
> The authors of Paper 4526

---

> ### Comment · Reviewer_rfd7 · 2025-08-06
>
> Thank authors for the detailed response. My concerns about technical details have been addressed. Although I still think adapting 2D technology to 3D is not novel enough, I will increase the score given the good performance number.

---

> ### Author Response · Authors · 2025-08-07
> **Thank Reviewer rfd7 for the increased score**
>
> We are grateful for your increased score. We will incorporate the additional results and make our novel aspects in comparison to the previous 2D technologies clearer following your suggestions. We hope our contributions along with the good performance can serve as a strong baseline and inspire more innovations to advance the transformer-based 3D perception community.

---

### Note · Authors · 2025-08-14

Dear ACs and Reviewers,

We sincerely thank you for your dedication in this review process and valuable suggestions. We are encouraged that the reviewers recognize SynCL's contributions in advancing the transformer-based 3D perception community:

* Rich experiments and clear analyses that reveal the optimization difficulties in joint detection and tracking training, focusing on attention mechanisms and bipartite matching.

* Good writing and rational design with Hybrid Matching for recovering over-deduplicated object queries, Dynamic Query filtering for uncovering high-quality 3D candidates, and Contrastive Learning to align these features with track queries and mitigate self-centric attention issues.

* Across various benchmarks, SynCL delivers consistent improvements and achieves new state-of-the-art performance on the nuScenes dataset, which is orthogonal to the OneTrack's joint optimization paradigm as shown in the rebuttal.

The reviewers' concerns are well addressed in the detailed response. For example, Reviewer `rfd7` decides to increase the score given SynCL's good performance, Reviewer `VTpN` maintains the initial weak accept rating and increases the Confidence score from 2 to 3 after checking our feedback, and Reviewer `A2ku` recognizes our efforts in providing an excellent rebuttal and decides to raise the rating to Accept.

We believe our contributions along with the good performance can serve as a strong baseline for 3D tracking and have important inspirational value for the transformer-based 3D perception community.

Best regards,

The authors of Paper 4526

---

### Decision · Program_Chairs · 2025-09-17

**Decision:**

Accept (poster)

**Comment:**

Paper Summary:
This paper identifies that the self-attention mechanism in tracking-by-attention models causes issues like over-deduplication for new object queries and self-centric attention for existing track queries. To address this, the authors propose SynCL, a plug-and-play training strategy that uses a parallel weight-shared decoder without self-attention, alongside several other learning components, to improve the joint learning of detection and tracking tasks.

Main Strengths and Weaknesses:
The primary strength is its strong empirical results; the proposed method is a well-engineered, plug-and-play solution that demonstrates significant performance improvements when applied to multiple state-of-the-art 3D tracking baselines. The main weakness is the limited technique novelty, as key components of the solution (e.g., a parallel decoder without self-attention) share strong similarities with prior work in 2D object detection and tracking.

Rebuttal Analysis:
The rebuttal addressed all technical questions and clarity concerns well, leading one reviewer (rfd7) to raise their score and another (A2ku) to affirm their high confidence after receiving detailed responses and new experiments. While some reviewers (FdMH, VTpN) maintained their scores, still harboring concerns about novelty, the rebuttal was strong enough to solidify a positive consensus.

Final Justification:
All four reviewers converged on a positive recommendation, ranging from "Borderline Accept" to "Accept." While the concerns about novelty remain, the practical contribution of this paper is significant. Considering the identification of a relevant problem in 3D multi-object tracking and the given effective solution, the AC agrees with the reviewers and recommends accepting this paper. Additionally, it is strongly suggested that the authors incorporate the details discussed in the rebuttal to make the paper more comprehensive.